# Sociodemographic Factors Influencing the Use of eHealth in People with Chronic Diseases

**DOI:** 10.3390/ijerph16040645

**Published:** 2019-02-21

**Authors:** Fabienne Reiners, Janienke Sturm, Lisette J.W. Bouw, Eveline J.M. Wouters

**Affiliations:** 1School of Allied Health Professions, Fontys University of Applied Science, P.O. Box 347, 5600 AH Eindhoven, The Netherlands; f.reiners@student.fontys.nl (F.R.); l.bouw@fontys.nl (L.J.W.B.); 2School of HRM and Psychology, Fontys University of Applied Science, P.O. Box 347, 5600 AH Eindhoven, The Netherlands; j.sturm@fontys.nl; 3Department of Tranzo, School of Social and Behavioral Sciences, Tilburg University, P.O. Box 90153, 5000 LE Tilburg, The Netherlands

**Keywords:** eHealth, digital divide, chronic diseases, sociodemographic factors, technology, interventions

## Abstract

Alongside the growing number of older persons, the prevalence of chronic diseases is increasing, leading to higher pressure on health care services. eHealth is considered a solution for better and more efficient health care. However, not every patient is able to use eHealth, for several reasons. This study aims to provide an overview of: (1) sociodemographic factors that influence the use of eHealth; and (2) suggest directions for interventions that will improve the use of eHealth in patients with chronic disease. A structured literature review of PubMed, ScienceDirect, Association for Computing Machinery Digital Library (ACMDL), and Cumulative Index to Nursing and Allied Health Literature (CINAHL) was conducted using four sets of keywords: “chronic disease”, “eHealth”, “factors”, and “suggested interventions”. Qualitative, quantitative, and mixed-method studies were included. Four researchers each assessed quality and extracted data. Twenty-two out of 1639 articles were included. Higher age and lower income, lower education, living alone, and living in rural areas were found to be associated with lower eHealth use. Ethnicity revealed mixed outcomes. Suggested solutions were personalized support, social support, use of different types of Internet devices to deliver eHealth, and involvement of patients in the development of eHealth interventions. It is concluded that eHealth is least used by persons who need it most. Tailored delivery of eHealth is recommended.

## 1. Introduction

Chronic diseases, also known as non-communicable diseases, constitute a significant challenge for healthcare. As reported by the WHO, each year 71% of deaths are due to chronic diseases [1]. Moreover, the number of people with chronic diseases is steadily growing when considering the increase in the (relative and absolute) number of older persons. At present, Europe has the highest percentage of people aged 60 or over [2]. In other parts of the world, rapid ageing will occur, such that a quarter of their populations will be older than 60 [2]. In addition, the economic burden of chronic diseases and the workload for health caregivers will increase accordingly [3,4,5]. 

The most prevalent chronic diseases are cancer, chronic respiratory disease, cardiovascular disease, and diabetes. All these diseases are to some extent related to behavioral risk factors such as tobacco use, unhealthy diet, lack of physical activity, and use of alcohol [6]. Thus, patients need to take responsibility in order to actively change their behavior and manage their chronic illness. This concept, which is known as self-management, is currently widely adopted to improve health outcomes and quality of life among chronically ill patients [7,8].

Nowadays, the rapid evolution of digital technology is leading to increased opportunities for self-management [9]. eHealth, for instance, provides the means to facilitate communication between care providers, and between care providers and their patients to exchange information and facilitate patients in self-monitoring [8], and is used to stimulate self-management in individuals with chronic disease [10,11,12,13]. For example, patients with chronic heart failure can monitor their own blood pressure and weight and communicate these data with their care provider.

In order to scale up the deployment of eHealth applications, especially for persons with chronic diseases needing health care, these applications should be easy to use for underserved patients, e.g., older persons with chronic diseases in lower socio-economic situations. The existence of groups of patients with chronic disease who are not able to benefit from eHealth, leads to a “digital divide” between those who use digital technology to self-manage their disease and those who do not use digital technology. 

As the use of technology in health care is increasing, it is essential to know sociodemographic characteristics such as age, sex, income, education, ethnicity, place of residence, and household composition of patients who are not aware or not able to use eHealth and what barriers these patients face [14,15]. Currently, an overview of such factors is missing. Additionally, no current review highlights the interventions (i.e., suggested solutions to enhance eHealth use) to overcome digital divide regarding sociodemographic factors. Therefore, this study aims firstly to give an overview of sociodemographic factors that influence the use of eHealth among people with chronic disease, and, secondly, the suggested directions for interventions to overcome these barriers and thus facilitate the use of eHealth for a broader population.

## 2. Materials and Methods 

### 2.1. Search Strategy

This literature review was conducted according to the Preferred Reporting Items for Systematic Reviews and Meta-Analyses (PRISMA) guidelines [16]. In November 2018, four databases were explored to conduct the research: PubMed, ScienceDirect, Computing Machinery Digital Library (ACMDL) and Cumulative Index to Nursing and Allied Health Literature (CINAHL). Groups of search terms, listed in Table 1, were entered into the four databases. Search terms from all four groups had to be present in the resulting article to be considered relevant. The precise combination of these groups of search terms can be found in the Appendix A (Combination of search terms according to the database used).

### 2.2. Article Selection

According to the PRISMA flow diagram, articles were first screened by title, then by abstract, and finally by full-text [16]. Articles were selected if they investigated sociodemographic factors influencing the use of eHealth for chronic diseases. Additional inclusion criteria were that articles were published in the last 10 years, articles were peer-reviewed, and that articles were written in English. Qualitative, quantitative as well as mixed-methods research articles were included. Review articles and theoretical papers were excluded. The target population selected were people with chronic diseases such as cardiovascular disease, diabetes, cancer, chronic respiratory disease, Alzheimer’s disease, dementia, and obesity.

### 2.3. Data Extraction

Four authors read the included articles. One author (F.R.) read all the articles, the other authors (L.B., J.S., and E.W.) each read a selection of the articles. Each author individually extracted the data into a data extraction form, which can be found in the Appendix A (Extraction Form). This form included information about the purpose of the study and the main outcomes, the target population and the technology, as well as the factors and suggestions for interventions to influence the use of eHealth. 

### 2.4. Assessment of Methodological Quality

All included articles were assessed for quality using the Mixed Methods Appraisal Tool (MMAT) [17]. The MMAT is a critical appraisal tool for the assessment of the quality of studies included in mixed studies reviews, including qualitative, quantitative and mixed-method studies. It appraises the quality of studies in five categories: qualitative research, randomized controlled trials, non-randomized studies, quantitative descriptive studies, and mixed-methods studies. For each category the MMAT provides a number of quality criteria that should be rated. 

### 2.5. Data Analysis

First, for each article, two of the authors had to reach agreement regarding the entry of the quality of each article and the data entered in the data extraction form. If there was no agreement, a third author assessed the article and the article was discussed until agreement was reached.

## 3. Results

The search in four different databases, relating to sociodemographic factors influencing the use of eHealth for chronic diseases, delivered 1639 results. In total, 166 articles were removed as they were identified as duplicates. After screening titles, abstracts, and full-texts on relevance for the research questions, 22 articles were included in the present literature review (Figure 1). Most of the full-text articles were excluded due to participants that were not diagnosed with a chronic disease but were at risk to have a chronic disease. Other reasons for exclusion were articles that investigated the use of Internet or technologies that were not related to eHealth or the eHealth technology was used to diagnose chronic disease.

### 3.1. Quality of Included Articles

After assessing the quality of the articles with the MMAT, most articles were found to be of mediocre quality. Frequently, the inclusion criteria for the participants’ selection required ownership of a smartphone, access to Internet connection, country language skills, and specific literacy skills. Moreover, in many studies confounders were not taken into account regarding the correlations found between sociodemographic characteristics of the participants and the use of eHealth. For example, people with higher education might also have a higher income. Other reasons for lower quality were small sample sizes and high risk of non-response bias, which decreased the generalizability of the results. A detailed table of the quality assessment of the included articles can be found in the Appendix A (Quality Assessment Table According to MMAT).

### 3.2. General Findings in the Selected Articles

The selected articles were screened for any sociodemographic factors influencing the use of eHealth among people with chronic diseases as well as for any suggestions for interventions to overcome these factors. The main findings are summarized in the Appendix A (Sociodemographic factors influencing the use of eHealth and suggested interventions). 

Articles were published from 2008 to 2018, with most of the articles being recent (2015–2018). Various study designs were among the included articles: one qualitative study, two randomized controlled trials, one mixed method, eleven quantitative nonrandomized studies, and seven quantitative descriptive studies (See: Appendix A (Sociodemographic factors influencing the use of eHealth and suggested interventions)).

The overall aim of the studies with respect to eHealth was diverse. Articles studied, for instance, the use of, the acceptance of, the access to, the effectiveness of, the willingness to use, the engagement in, and the interest in eHealth. Nevertheless, most often, the use or intended use of eHealth was investigated. The selected articles were from different geographical regions that represent various cultures. Also, a variety of technologies was used: six articles used mobile phones, one used a web portal, and one an Internet-based program. Most articles, however, did not report a specific form of eHealth technology. Articles that did not report a specific form of eHealth technology often described the results of a survey among people with a chronic disease in which they were asked about their use of eHealth in general. 

The selected articles also showed diversity of chronic diseases that were addressed; seven articles reported on patients with diabetes, four on cancer, four on heart disease, two on respiratory disease, one on dementia and four articles reported on patients with chronic disease in general. 

### 3.3. Sociodemographic Factors

#### 3.3.1. Age

The majority of the studies (*n* = 13) indicate age to be a significant factor that influences the use of eHealth [18,19,20,21,22,23,24,25,26,27,28,29,30]. Younger people are more willing and have more experience and interest in using eHealth compared to older people [18,20,21,22,24,26,27,28,29]. Often, the problem mentioned is that older persons do not own a device with which they can access eHealth, or, if they do own such a device, do not have the skills required to use it [18,20,21,29]. Also, the younger population is more aware of eHealth possibilities compared to older persons who have less knowledge about eHealth. Terschüren et al. report that older people are afraid of losing personal contact with their physician when they start using eHealth [30]. On the other hand, in the study of Goyal et al., it is shown that older persons, once they adopt the technology, adhere longer to a risk management app than younger people who, although they download the app more often, also lose interest more often [19]. In contrast, five studies indicate age not to be a significant influencing factor for the use of eHealth [31,32,33,34,35].

#### 3.3.2. Gender

The results regarding gender are not consistent. Only five out of 22 articles studies show that gender is a factor influencing the use of eHealth [19,20,30,31,36], whereas in eight studies no relation is found between gender and the use of eHealth [21,22,23,24,25,29,32,33]. In three studies, women are found to be more engaged and satisfied with eHealth applications and use them more often than men [19,36,37]. In contrast, the studies by Terschüren et al. and Kamis et al. show men to be more likely to accept telemonitoring than women [20,30].

#### 3.3.3. Income

Five articles addressed the relation between the income of the patients and the use of eHealth [24,26,31,35,37]. People with higher income tended to have more interest in eHealth compared to people with lower income [26,31]. It is indicated that lower income is associated with limited availability and access to Internet healthcare resources [31,37]. On the other hand, Whittemore et al. found that in people with lower income, the satisfaction in using eHealth is as high as for people with higher income [31]. In the studies of Nelson et al. and Song et al. income was not a factor influencing the engagement in eHealth [24,35]. 

#### 3.3.4. Education

Three studies indicate that education is not associated with the use of eHealth [28,34,35]. However, six studies indicate that education does play a role in the use of eHealth in chronic disease. These articles show that higher education is correlated with more knowledge as well as more use of eHealth technologies [18,20,26,30,33,38]. People who are educated to a higher level are more interested and more experienced in using eHealth [18,38]. Also, a lower level of education is associated with less adherence to telehealth [33].

#### 3.3.5. Vocational Status

Only two of the included articles investigate the relation between vocational status of people with chronic disease and their use of eHealth [26,38]. LaMonica et al. show that whether people are employed or not does not influence the use of eHealth [38]. In the study of Drewes et al., the same result is given [28]. 

#### 3.3.6. Ethnicity

Four articles report no correlation between eHealth use and ethnicity [23,26,34,35]. Whittemore et al. indicated that black, Hispanic, or mixed race/ethnicity are more likely to refuse to enroll in Internet-based research compared to white adolescents [31]. Also, it was suggested that non-white participants experienced pressure of the healthcare provider to enroll. Once they were enrolled in the research though, the satisfaction was as high as for white participants [31]. Suggested reasons for refusal are problems with Internet access or financial issues that lead to a lack of Internet availability [31]. Nelson et al. found that race is significantly related to text responses in eHealth. Non-white participants are less likely to respond to text messages compared to white participants. Similar results were found for participation in interactive voice response calls [24]. 

#### 3.3.7. Place of Residence

Six articles reported that people with chronic diseases living in rural areas are less likely to use eHealth compared to urban citizens [22,23,29,32,37,39]. They were found to have less access to eHealth, and the implementation of eHealth is less effective. As assessed by Duplaga et al. the use of eHealth is more common in urban areas with more than 100,000 inhabitants [22]. As explained by Han et al., the geographical influence is often related to lower socioeconomic status [37]. A positive effect of eHealth was seen in South Korea. A city with few medical facilities showed greater use of eHealth compared to an urban city [32]. Contrarily, Terschüren et al. showed that the place of residence does not play a role in the use of telemedical devices among people with chronic diseases [30].

### 3.4. Suggested Directions for Interventions

#### 3.4.1. Personalization of eHealth; Different Ways of Information Delivery

Taking into account all sociodemographic factors that have been shown to influence the use of eHealth (i.e., age, income, education, living alone, place of residence, and ethnicity) some articles highlight that eHealth interventions need to be tailored to the specific target population [30]. An eHealth strategy should not be “generalized” for all patients but needs to be adapted to the patient’s profile. Offering these customizations might enhance acceptance and motivation of eHealth use [22,24,28,37,38]. 

Several studies show that lower educated people and older people need more support for being able to use eHealth, because they experience more problems with accessing and understanding eHealth information [18,25,27,34,38]. One suggestion is to include tutorials in the eHealth application [25]. Again, it is important that this support is adapted to the capability of the patients as well as their personal preferences [38]. Educational initiatives are suggested as a way to improve the perception and acceptance of eHealth by patients. Additionally, involving caregivers could enhance familiarization with the technology for novice eHealth users [21].

One of the reasons for low use of eHealth among older persons is their concern about missing immediate understandable feedback when they do not have personal contact with the health care provider [30]. Currently, people trust their general practitioner as an interpreter of measurement values in telemonitoring applications [30]. Also, they want to be sure of adequate help in case of acute or emergency conditions [30]. Informing people on how this works in a telemonitoring application may help to improve acceptance among older adults [30]. In addition, eHealth awareness can be enhanced by using media older persons are familiar with; in the same study it was found that magazines and television were valuable as a first step to implement eHealth strategies [30].

Nelson et al. confirmed that eHealth should also be tailored to cultural attitudes and beliefs to increase engagement [24]. As a manner to achieve this goal, Anglada-Martinez et al. suggested improving the perceptions of the potential eHealth users by offering examples in which people can better recognize themselves [21]. They suggest, for instance, to include movie material and other creative approaches with people representing the same race to reach ethnical minorities [31].

The ease of use of eHealth strategies is indicated to be helpful to increase motivation and use of eHealth. EHealth interventions should save time, contain a reasonable amount of information and be comprehensible [31,39]. Therefore, it is essential to include the target population in the design and development of the eHealth tool [24,31]. In this way, the eHealth intervention can be used more effectively [39], and ethnical disparities could be avoided [31].

#### 3.4.2. Facilitate Access to Internet; Use of Different Devices and Modes of Delivery

Frequently, mobile phones are used for eHealth interventions [27]. Mobile phones are common even among lower income populations and are cost effective. For example, texting is easy and can provide reminders and key tips to the patients [38]. Smith et al. reported that text messages can provide lifestyle advice and health awareness. Websites should be mobile-friendly, to avoid excluding people without computers or tablets [38]. Multiple platforms should be offered in order to include youth with diverse race as well. For example, Latino and black youth are more likely to access the Internet by cell phone compared to white youth who are more likely to use a computer [30]. 

In contrast, it was shown by LaMonica et al. that older people prefer using computers compared to mobile phones, and therefore using the mobile phone as the only option still leads to disparities in eHealth [38].

Social media is getting more popular and is nowadays, amongst others, considered as a potential eHealth strategy [38]. Engagement in eHealth by diverse youth with diabetes can also be successful by using social media including interactive blogging, connecting with others and creative expressions [30]. It was also suggested that the use and uptake of social media might be fundamental for interventions that target older adults having cognition issues [38].

A big issue among eHealth interventions is the access to the Internet [26]. Easy access could be provided by giving eHealth options before and after an appointment in the clinic or health facility or even during the treatment session [30,35]. Other suggestions are to subsidize the costs of computer devices, as technology is found to be costly [26]. Furthermore, eHealth should be available offline once it is downloaded, especially for patients living in rural areas [39]. 

#### 3.4.3. Inclusion of Family Members

According to a number of articles, family members play an essential role in the implementation and use of eHealth [24,35,38]. For instance, older persons who experience difficulties with technology often are helped by family members [24,35,38]. Besides, multiple person households positively influence the use of eHealth [28]. Family members can also help patients to accept eHealth by encouraging them, especially helping them in case of low eHealth literacy [35].

This also counts for younger eHealth users. Whittemore et al. showed that parents can be the key to stimulate young diabetes patients to use eHealth. Especially for African American, black, Hispanic or mixed-race youth, interest in eHealth can be enhanced through their parents. Therefore, health care providers need to know how to convince parents to have their adolescents use eHealth. This could be done by personal interviews or skype-meetings with the parents. Moreover, once these persons use eHealth, they show high satisfaction; therefore creative recruitment strategies are needed [31]. 

#### 3.4.4. Complementary Remarks Considering Improvement of Uptake of eHealth

Another option to encourage patients to use eHealth, according to Jacobs et al., is to persuade patients with the argument that by using eHealth and sharing their data, they can help other patients with the same issues [39]. Others reported that in some cases a phone call may be a better way to communicate with a patient than sending a text message, since in some parts of Latin America, people can receive a mobile phone call free of charge [39]. 

In general, several articles recommend to use eHealth as a complement rather than replacing current health delivery systems [19,26,27,29,31,36]. People not owning any technical devices to access health information should not be excluded [27]. It is argued that reaching acceptance of eHealth among people with a chronic disease is easier if it is used as a complementary health care delivery system [27,29].

## 4. Discussion

### 4.1. Summary of the Main Results

This review identified and synthesized the literature regarding sociodemographic factors influencing the use of eHealth in people with chronic diseases. It also addressed suggestions to facilitate or improve the use of eHealth. Findings suggest that sociodemographic factors influencing the use of eHealth are diverse and complex. All the included articles reported several factors acting as barriers or facilitators in the use of eHealth. Of all the factors, age has been studied most extensively; it was addressed in 18 out of 22 included articles [18,19,20,21,22,23,24,25,26,28,29,30,38]. Vocational status was addressed in only two of the articles [28,38]. 

Older persons and persons with lower income and/or lower education are less likely to use eHealth. Also, people with chronic diseases who are living alone are less inclined to use eHealth, because family members often help with difficulties experienced while using eHealth. Marital status has not been addressed explicitly in any of the studies. Vocational status has been shown not to influence the use of eHealth.

Regarding gender, the results did not show consistency. Ethnicity plays an important role in the one article that studied ethnicity: non-white people used eHealth less compared to Caucasians. Nevertheless, they are more satisfied once enrolled in an eHealth program. 

A noteworthy factor influencing the use of eHealth is the place of residence. People living in rural areas have less access and opportunities regarding eHealth. However, these are the people most in need of eHealth care to overcome scarcity in health care provisions in these distant areas. It was striking that in only one study eHealth was found to be a facilitator for people living in rural areas because they had fewer medical facilities and longer travel time. In addition, people living in rural regions have a lower socioeconomic status, which is another barrier. 

In many of the articles included in the present literature review suggestions were made to improve the use of eHealth, which can be clustered in three main themes. First, eHealth applications should be personalized and tailored to the specific target population. For instance, older persons get to know eHealth through magazines and television; therefore, these are important media for the first step of implementation. Also, older persons, as compared to younger ones, need direct feedback from data that result from monitoring their vital systems, and are willing to get assistance from qualified practice assistants instead of the general practitioner. Among people with chronic diseases, eHealth should be easy to use, with short texts and visual material. Another upcoming eHealth tool is social media, which might be appealing for young people. The involvement of the patients in the designing and developing part is essential. 

Second, access to Internet should be facilitated and made as easy as possible. For instance, for low income groups, it is suggested that mobile phones should be used, as communicating using a text message is easy and cheap. Others argue that communicating by making a phone call is a better option, because making a phone call does not require literacy skills and might be suitable for less educated people. In addition, websites should be mobile-friendly. Multiple platforms should be available for eHealth interventions to avoid disparities, because, e.g., older persons prefer computers over mobile phones. Regarding access to the Internet, eHealth should be available before, during and after clinical appointments. Besides, eHealth should be available offline once it has been downloaded. In general, eHealth should be used as a complement to avoid health inequalities.

Third, family members should play a role in the use of eHealth. Older people and lower educated people need more support when using eHealth technologies. Involving family members can help to resolve issues with eHealth literacy and lack of technology skills. One article suggested to involve parents to increase the interest in eHealth among young diabetes patients. This finding also implies that, in situations that no family members are available, other human support might be necessary, e.g., neighbors, or volunteers. 

The results of this review underline previous observations that various forms or stages of digital divide can be distinguished [40]. First, there is the economic divide, manifested in the fact that not everyone can afford to buy the hardware needed to access eHealth. This is influenced by factors such as income, vocational status and place of residence. Second is the so-called usability divide, meaning that some people may not be able to achieve the benefits of eHealth because it is too difficult for them to understand and use effectively. This is applicable to for instance illiterate people or older people with decreasing cognitive and physical skills, and thus influenced by factors like age and education. The third and final form is the empowerment divide, which related to the fact that although they have access to the right hardware, and they have the skills to use the eHealth applications, some people will not make use of the opportunities offered by eHealth, because they do not feel personally empowered to do so and do not think they will benefit. The empowerment divide is much more complicated to overcome. In their suggestions for interventions, the articles included in this literature review therefore mostly address the economic and the usability divide. Therefore, in future research, the empowerment divide deserves specific attention. 

### 4.2. Strengths and Limitations 

To our knowledge, this is the first review to address sociodemographic factors as related to the use of eHealth in chronic diseases. Also, it is the first attempt to summarize possible solutions to overcome the digital divide in the use of eHealth, as suggested in the existent research. Therefore, it offers suggestions to improve the adoption and use of eHealth applications in practice. A clear methodological approach was used to conduct the research validated by several authors comparing their findings. Another strength of this literature review is the use of multiple databases to conduct the article selection. The quality of the articles was assessed by the MMAT, an effective tool to evaluate the quality of articles with diverse study types. A favorable aspect of this review is the fact that the studies are conducted in diverse countries with various chronic diseases, which increases the generalizability of the results. As the development of technology is growing fast, it is to be noted that the studies included are mostly recent. 

At the same time, several limitations need to be mentioned. As for sociodemographic factors associated to eHealth use, most of the researches used univariate statistics, resulting in bias by confounding. Hence, it is hard to conclude whether a specific factor influences the use of eHealth. In addition, the aims of the studies differed: some studies evaluated to what extent eHealth was actuallyused by patients, whereas most studies described the acceptance and future use of eHealth. A major limitation of this review is that the interventions suggested in the literature have not been empirically tested. Therefore, future research should evaluate the effectiveness of such interventions in practice. 

## 5. Conclusions

Research demonstrates that eHealth is least used by persons who appear to need it most, i.e., older persons, persons with chronic diseases, persons in rural areas, and persons with lower levels of education. In order to scale the use of eHealth, it is necessary to address this inequality. Suggested directions for interventions are to personalize the content and mode of delivery of eHealth and to design together with the intended users. Also, the choice of the used Internet devices should be able to vary according to familiarity and preference of the user. Finally, eHealth should be accompanied by human support.

## Figures and Tables

**Figure 1 ijerph-16-00645-f001:**
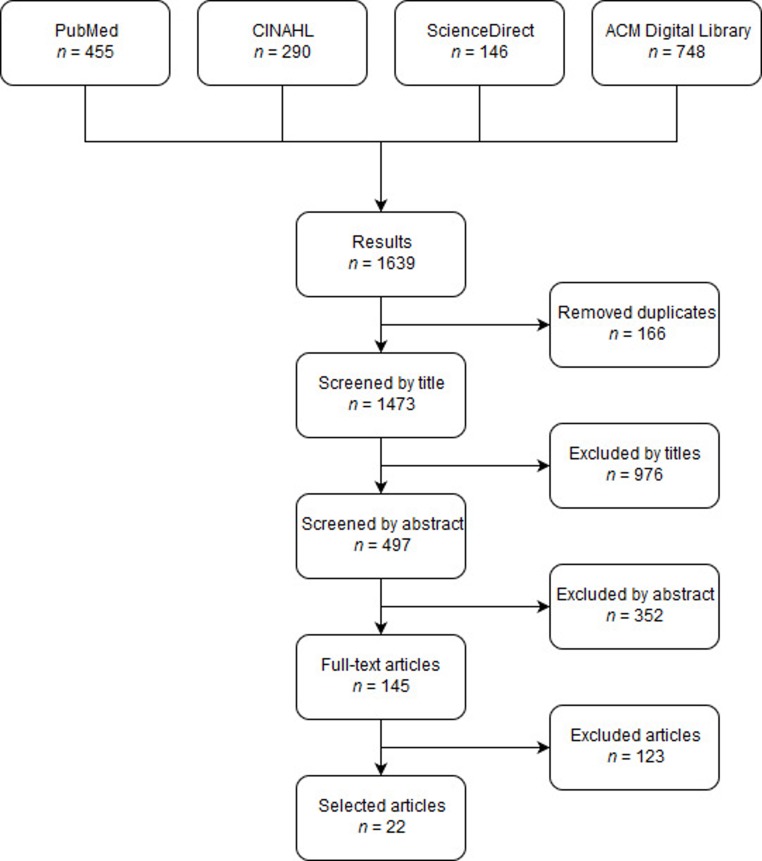
Flow diagram of the article selection process.

**Table 1 ijerph-16-00645-t001:** Groups of keywords.

Group	Search Terms
(1) Chronic disease	“chronic diseases” OR dementia OR “Alzheimer’s disease” OR obesity OR “non-communicable disease” OR NCD OR “non-communicable condition” OR “cardiovascular disease” OR diabetes OR cancer OR “chronic respiratory disease” OR “chronic condition” OR “lung disease” OR “heart disease” OR “long-term disease” OR hypertension OR “chronic illness”
(2) eHealth	e-Health OR eHealth OR “electronic health” OR “electronic devices” OR “Web 2.0” OR “net health” OR “digital health technology” OR “health care information” OR “interactive health communication” OR telemonitoring OR tele-health OR mHealth OR telehealth OR telecare OR “health technology”
(3) Factors	factors OR barriers OR characteristics OR facilitators OR “sociodemographic factors” OR predictors OR “digital divide” OR “health inequalities” OR disparities OR inequalities
(4) Suggested interventions	programme OR recommendations OR recommendation OR program OR strategy OR suggestion OR implementation OR adoption OR trial OR solution OR coaching OR intervention

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
