# Peer review of "Sociodemographic Factors Influencing the Use of eHealth in People with Chronic Diseases"

_ijerph, 2019, doi:10.3390/ijerph16040645_

Round 1

Reviewer 1 Report

It is a good job of systematic review of literature. Well defined Deataco the section of strengths and weaknesses that is very illustrative, and usualli they are not included. 

The work would be clearer if they included the appendices, which only quotes them.

Author Response

Response to Reviewer 1 Comments

Point 1: It is a good job of systematic review of literature. Well defined Deataco the section of strengths and weaknesses that is very illustrative, and usualli they are not included. 

Response 1: We thank the reviewer for the compliments on our work!

Point 2: The work would be clearer if they included the appendices, which only quotes them.

Response 2:

- Appendix A is, as advised, we intended to add in the manuscript as ‘table 2’.  After doing so and after revising the manuscript as suggested by all reviewers, we revised again to the original tables. After discussion, we found that the adding of Appendix A was not adding enough to the paper and to table 1 to include it.

- Appendix B is the (original) MMAT extraction form, which can be informative for other authors who intend to do a mixed method literature review. We did not add that to the manuscript, because it is not specific for our work.

- Appendix C is the original quality assessment form. Because we described the quality of the articles rather extensively in the text of 3.2, we have decided, after discussion between the authors, that it is not of great value to add this in the text. The more, as the MMAT group advises to use the quality assessment in a descriptive, rather than a ‘quantitative’ manner.

We did, however, change the text in the Method and especially, in the result section, and extended some of the descriptions, in a way that the Appendices are not missing for clarity (see original manuscript and our changes).

Reviewer 2 Report

Well  done, this literature review provides a current synthesis of the participation in eHealth of people with chronic diseases and the impact of the social determinants of health.

I have made considerable constructive suggestions throughout the article (uploaded). However, I do not believe these should deter you from making revisions and ultimately publishing an informative article.

Please also ensure your 'extraction form' is correctly headed as an appendix (I have not uploaded this form)

Thank you for allowing me to peer review your article.

Author Response

Response to Reviewer 2 Comments

Point 1:

Well  done, this literature review provides a current synthesis of the participation in eHealth of people with chronic diseases and the impact of the social determinants of health.

Response 1: We thank the reviewer for the compliments on our work!

Point 2:

I have made considerable constructive suggestions throughout the article (uploaded). However, I do not believe these should deter you from making revisions and ultimately publishing an informative article.

Response 2: We appreciate the great effort you took to improve or work. We followed your comments as follows:

Point 2.1: Throughout the article the words ‘participation’ in eHealth and ‘use’ of eHealth have been used interchangeably. As the word ‘participation’ is included in the title, the article would benefit from a definition of ‘participation’ and then consistent use of this word

Response 2.1: we agree with the reviewer that our use of the word ‘participation’ is inconsistent and was used interchangeably with the word ‘use’. Reconsidering this, we believe that in most cases (and articles included) to use the word ‘use’ would be more appropriate. Therefore, we changed ‘participation’ (also in the title) into ‘use’ in most cases.

Point 2.2: Line 1 refers to a ‘structured literature review’ Line 18 refers to a ‘systematic review’

Response 2.1: we changed the wording in line 1 into ‘systematic review’.

Point 2.3: Requires reorder to reflect the Title. For example, sociodemographic factors, followed by eHealth, followed by people with chronic disease

Response 2.3: The authors discussed this suggestion and finally agreed that it is most logical to keep the order like it is, to best introduce the topic even if it is not following the order of the title.

Point 2.4: ‘Reword clumsy’

Response 2.4: we changed ‘especially for those most in need of chronic health care’ into: ‘especially for persons with chronic diseases, needing health care’.

Point 2.5: Define usable.

Response 2.5: we changed this into: ‘…these applications should be easy to use by…’.

Point 2.6: This paragraph requires citations literature is available to support this comment Showell, C., 2017. Barriers to the use of personal health records by patients: a structured review. PeerJ, 5, p.e3268.

Response 2.6: Thank you for offering us this reference. We added it in the paper.

Point 2.7: Define (sociodemographic characteristics)

Response 2.7: we added in the text: ….sociodemographic characteristics, such as age, sex, income, education, ethnicity, place of residence and household composition…’

Point 2.8: Move line 126 ‘article published between… to here’

Response 2.8: We chose not to move the line from the Results section into the Method section, as it was not an inclusion criterion, but an actual result. Instead, in paragraph 2.2 ‘Article selection’, we wrote that we included articles that were published over the last ten years.

Point 2.9: Please correct

Response 2.9: Throughout the paper the incorrect citations were corrected, and missing citations were added.

Point 2.10: is to à can

Response 2.10: Throughout the paper all typing and language errors were corrected.

Point 2.11: should the individual factors have been included in the search? If not why not?

Response 2.11: we have chosen to define sociodemographic as broad as possible. Therefore, we included ‘factors’ and ‘barriers’ as well as sociodemographic factors. It might be possible that we would have been able to find specific associations by including all individual sociodemographic factors but on the other hand, we decided that, by also using ‘factor’ as a search term, we would have a maximum of hits. Especially, as studies used different words for these factors.

Point 2.12: reword ‘a pair of two authors’

Response 2.12: we changed this into: ‘two of the authors’

Point 2.13: remove ‘relatively’

Response 2.13: we removed ‘relatively’ as suggested by the reviewer.

Point 2.14: ‘all factors (sociodemographic factors)’?

Response 2.14: because we use (also) broad terms to find associations with eHealth use, we expected to find ‘all’ sociodemographic factors. In the studies included, several factors were referred to with varying wordings. We changed ‘sociodemographic factors’ into ‘any sociodemographic factors’.

Point 2.15: Capitals in table 2

Response 2.15: thank you very much for your thoroughness. We changed into Capitals where appropriate.

Point 2.17:  Respiratory, be consistent

Response 2.17: we changed ‘lung’ into ‘respiratory’.

Point 2.19: So what does this mean for participation in eHealth? All research?

What does this mean? This could be an interesting finding?

Response 2.19: The findings refer to the reference mentioned, i.e., only one study. By changing the sentence from present to past tense, we made it clearer that this finding refers to one study, and not to all research. We also rephrased this paragraph as indicated in the manuscript (with ‘track changes’).

Point 2.20: remove SES

Response 2.20: we removed SES as suggested

Point 2.21: ‘This section requires revision, strengthening and consideration of use of language’ (section 3.4)

Response 2.21: we revised the section as suggested, adding citations. The last paragraph was changed into:  The ease of the use of eHealth strategies is indicated to be helpful to increase motivation and engagement in use of eHealth. EHealth interventions should save time, contain a reasonable amount of information and be comprehensible[32,40]. Therefore, it is essential to include the target population in the design and development of the eHealth tool [25,32]. In this way, the eHealth intervention can be used more effectively completion rates are higher [40], and , as mentioned by Whittemore, ethnical disparities could be avoided [32].”

Point 2.22: Which factors influence participation in eHealth?

Response 2.22: we added the specific factors influencing use of eHealth.

Point 2.24: professional and/or family?

Response 2.24: We chose to leave the word caregivers unchanged, because from the original paper it is not clear whether they refer to professional caregivers or family or both.

Point 2.27: Change wording of ‘In the same line of thought’

Response 2.27: We left out this part of the sentence.

Point 2.28: Part of sentence is used repeatedly

Response 2.28: Whenever sentences were used repeatedly, this was corrected.   

Point 2.29: ‘Completion rates are higher’ what does this mean?

Response 2.29: We changed this sentence into ‘Thus, the eHealth intervention can be used more effectively’.

Point 2.30: ‘But section 3.3.6 challenges the use of texting among ethnic populations, be clear in what your saying’, ‘As with comment above’

Response 2.30: We added a sentence specifically addressing this issue.

Point 2.31: ‘Are there ethical consideration here?’ in 3.4.4. and 3.4.5.

Response 2.31:

No, in the articles referred to, there are no ethical considerations mentioned.

Point 2.32: ‘that calling might be a better option’ What does this mean?

Response 2.32: We rephrased the sentence to make it more clear.

Point 2.34: ‘All or older people or people with chronic conditions?’

Response 2.32: we changed the wording into “people with chronic diseases living alone…”.

Point 2.35: ‘So what does this mean for eHealth?’

Response 2.35: We decided it was better to remove this sentence, because this issue has not been addressed explicitly in the Results section.

Point 2.36: What do you mean? (texts and movie material, as well as interactive games)

Response 2.36: we reworded the sentence to make it more clear.

Point 2.37: Use and usability have two different meanings in the context of eHealth. If used together they need clear concise explanation.

Response 2.37: We changed the wording of the sentence to “Second is the so-called usability divide, meaning that some people may not be able to achieve the benefits of eHealth because it is too difficult for them to understand and use effectively”.

Point 3

Please also ensure your 'extraction form' is correctly headed as an appendix (I have not uploaded this form)

Thank you for allowing me to peer review your article.

Response 3: Thank you for your attentiveness. We have changed the heading of the Excel data extraction for into: Appendix B: Extraction Form. Thank you very much for your very kind compliments!
